# The work of farmers in short food supply chains: Systematic literature review and research agenda

**Philippine Dupé**[1,2*], **Benoît Dedieu**[1], **Pierre Gasselin**[2], **Guillaume Ollivier**[3]

**1** UMR Selmet, University of Montpellier, CIRAD, INRAE, Institut Agro, Montpellier, France, **2** UMR Innovation, University of Montpellier, CIRAD, INRAE, Institut Agro, Montpellier, France, **3** UR Ecodéveloppement, INRAE, Avignon, France

* philippine.dupe@inrae.fr

## Abstract

### Background

Over the last 20 years or so, farmers the world over have been expressing renewed interest in Short Food Supply Chains (SFSCs). Since these marketing channels bring consumers and producers closer together, they are being promoted as a means for producers to recover more of the added value. They are also seen as a part of the solution to the problems associated with long supply chains. However, marketing via SFSCs means that farmers have to take on new tasks, which are likely to disrupt their existing work routines. We propose here to review the scientific knowledge available on the work of farmers in SFSCs.

### Methods

We carry out a systematic analysis of the English-language literature using a multidisciplinary analytical framework of work. We consider five dimensions of work: 1) structural elements of work; 2) economic performance of work; 3) organization of work; 4) working conditions; and 5) occupation.

### Results

The theme of work is not prominent in the existing literature on SFSCs. While there is ongoing debate whether a switch to SFSCs results in increased farm incomes, it is clear that marketing through SFSCs relies on a large amount of poorly remunerated work. New tasks have to be added to work schedules, and new skills are required, making the organization of both productive and marketing work more complex. Various categories of workers are involved in undertaking these tasks, having to bear a heavy workload that is a source of stress and fatigue. However, the economic aspect aside, SFSCs appear to be conducive to farmers' self-fulfillment at work. Farmers derive particular satisfaction from their involvement in SFSCs since they are more in touch with consumers and because they are participating in the transition of food systems, despite the intellectually and physically demanding work they have to do.

**Data availability statement:** All relevant data are within the manuscript and its Supporting Information files.

**Funding:** This work was financially supported by the Fondation de France and by the metaprogram "Métabio" from the National Research Institute for Agriculture, Food and the Environment (INRAE, France), which financed a PhD grant for Philippine Dupé. The funders had no role in study design, data collection and analysis, decision to publish, or preparation of the manuscript.

**Competing interests:** The authors have declared that no competing interests exist.

## Conclusion

We call for the study of farmers' work in SFSCs to be strengthened, by diversifying both the methodologies adopted and the case studies. Such knowledge should enable us to better integrate work-related issues into future public policies to support SFSCs.

## 1. Introduction

There has been a revival since the early 2000s of Short Food Supply Chains (SFSCs) in many countries [1–3,4]. Some SFSCs have existed for a long time (open-air markets, farm-gate sales), while others are more recent (teikei, Community Supported Agriculture (CSA), farmers' drives, online sales, collective catering, etc.). Although in some countries, the volumes of production sold through these marketing channels are not very significant, these SFSCs are now attracting a growing number of farmers [5,6]. Encouraging a reconnection between producers and consumers, or leading to a better distribution of added value, SFSCs are seen as part of the solution to certain problems caused by the dominant agricultural model [7–9,4]. They are therefore also gradually gaining institutional recognition and dedicated support [5,6,10].

Farm work related to marketing through SFSCs has been identified as a challenge [11–13] and as a major concern for farmers in SFSCs [12]. Since it is addressed by several disciplines [14], the notion of "work in agriculture" is multifaceted [15]. For farmers, marketing via SFSCs entails taking on tasks of marketing and even processing of products. These tasks, additional to those of their usual production activities, represent an organizational challenge [16]. They require specific non-agricultural skills, which can upend the very conceptions of the farming profession [17]. Based on elements of geographical or organizational proximity [18], SFSCs also help bring producers and consumers closer together [19], and contribute to a better distribution of added value for farmers [9,20,21,4]. These elements are likely to have an impact not only on the meaning of work for farmers, but also on a farm's economic performance [22]. While expectations regarding work are still evolving [23], the balance between working conditions and a sense of fulfillment in work perceived as meaningful is a topic of lively discussion [24,25]. Harsh working conditions are leading some farmers to leave the profession despite their attachment to an activity that contributes to an alternative agricultural and food model [26].

Some literature reviews have examined certain aspects of the work of farmers [12,27–29]. However, work is not at the core of these studies. That is why, in this review, we identify and characterize the knowledge available in the scientific literature on the work of farmers marketing via SFSCs. Using a systematic literature review method [30], we explore several aspects of agricultural work and several spatial and socio-organizational scales and consider a wide variety of SFSCs.

## 2. Method: systematic literature review

### 2.1. Definition of concepts

We adopt a broad and inclusive definition of SFSCs, encompassing modes of marketing based on various registers of proximity [18]. We thus cover diverse working situations that participate in supply chains that are "intermediated" or "direct", that involve individual farmers or collectives of them, that are "locally" anchored or not, etc. Other related concepts were also used to collect the articles ("Local Food Systems" (LFS) [8], "Alternative Food Networks" (AFN) [27,31] "Sustainable food supply chains" [28,32]) (S1). In this body of literature, we have taken care to retain only those articles that clearly refer to SFSCs.

We created an analytical grid around different dimensions of the "work of farmers" in SFSCs (S1). We mobilized five major prisms of analysis of agricultural work in SFSCs to draw up our query and to analyze the data stemming from the articles: 1) structural elements of agricultural work in SFSCs, at the levels of workers, farms and territories; 2) economic performance of agricultural work in SFSCs; 3) organization of the work of farmers in SFSCs; 4) working conditions of farmers in SFSCs (physical and mental hardship, motivations and satisfactions); and 5) new professions associated with the work of farmers in SFSCs.

## 2.2. Bibliographic search strategy

We used a Systematic Literature Review (SLR) method to identify existing scientific knowledge on the work of farmers marketing via SFSCs. SLR is a "predetermined structured method to search, screen, select, appraise and summarize study findings to answer a narrowly focused research question"[33]. By publishing a search protocol before the literature review (S1 Document), we have ensured the transparency and reproducibility of the search [34]. Our query is based on the three main concepts of our research question: work, agriculture and SFSC. We queried two databases, Web of Science Core Collection (WoS) and Scopus, according to the search criteria listed in Table 1.

## 2.3. Selection process

When we used our protocol (S1 Document) to query the databases, we obtained 789 articles (345 from WoS, 444 from Scopus). After eliminating duplicates, we were left with 494 articles. To select only the relevant publications from this initial corpus, we defined "eligibility criteria" (Table 2) using the PICO method (Participants, Intervention, Comparators, Outcome) [35]. These criteria, listed in Table 2, allowed us to retain only those publications that actually provide results on the work of farmers marketing via SFSCs. In the first stage, called "screening", we pre-selected 102 publications on the basis of their titles, abstracts and keywords. Screening was carried out by two of the review's authors, in double-blind conditions. The second phase, called "eligibility", consisted of full readings of the pre-selected articles to confirm their contribution to the subject according to our eligibility criteria (Table 2). These two phases led to 52 articles being left in the final corpus (Fig 1). A "citation chasing" stage, based on the examination of bibliographic references citing or cited more than twice by these 52 articles, allowed us to add 27 more articles, selected according to the same screening and eligibility conditions from the 426 articles citing or cited more than twice by our previously constituted corpus (Fig 1). Our final corpus thus consisted of 79 articles.

**Table 1. Search criteria for publications in the WoS and Scopus databases.**

| Reminder of search criteria |
| --- |
| **Thesaurus that has to be present in title, summary or keywords** |
| **W** = (work* OR labor* OR labour* OR job* OR task* OR employment* OR occupation*) |
| **A** = (agricultur* OR farm* OR rural*) |
| **S** = ("short* food supply*" OR "short* food chain*" OR "local* food system*" OR "local* food chain*" OR "alternativ* food network*" OR "alternativ* food system*" OR "sustainab* food supply*") |
| **Characteristics of the publications** |
| **Type of document:** Peer-reviewed articles and book chapters |
| **Date:** until 31/08/2024 |
| **Language:** English |

**Table 2. PICO table summarizing our publication eligibility criteria for the review.**

| PICO Components | |
|---|---|
| Participants (population) | Workers, farms, agricultural models, territories (links between farmers and other territorial actors necessary to carry out SFSC-related agricultural work) |
| Intervention (exposure) | Marketing of at least part of production via SFSCs |
| Comparators (control) | Farms that are not, not yet, or no longer marketing via SFSCs |
| Outcomes | - **Structural elements of SFSC work (Structure): associated categories of workers, farm structures and labor markets:** type of workers involved in agricultural work in short supply chains, production systems and activity systems of the farms concerned, labor market and employment rate linked to SFSC-related agricultural work at the farm and territorial scales.<br>- **Economic performance:** main economic indicators pertaining to work performance (productivity, profitability, income, etc.).<br>- **Work organization:** spatio-temporal organization of tasks between workers and social relations of production at the scale of: the farm; collective sales outlets; territories.<br>- **Working Conditions:** physical and mental hardship; meaning of work (any form of subjectivity and emotion in the work (stress, satisfaction, recognition, autonomy, sense of coherence, identity, etc.) or motivational regime around this work).<br>- **Occupation:** identity of the profession, skills (objectified, situational) and training, professional norms and values, practices, objectives, occupational solidarity. |

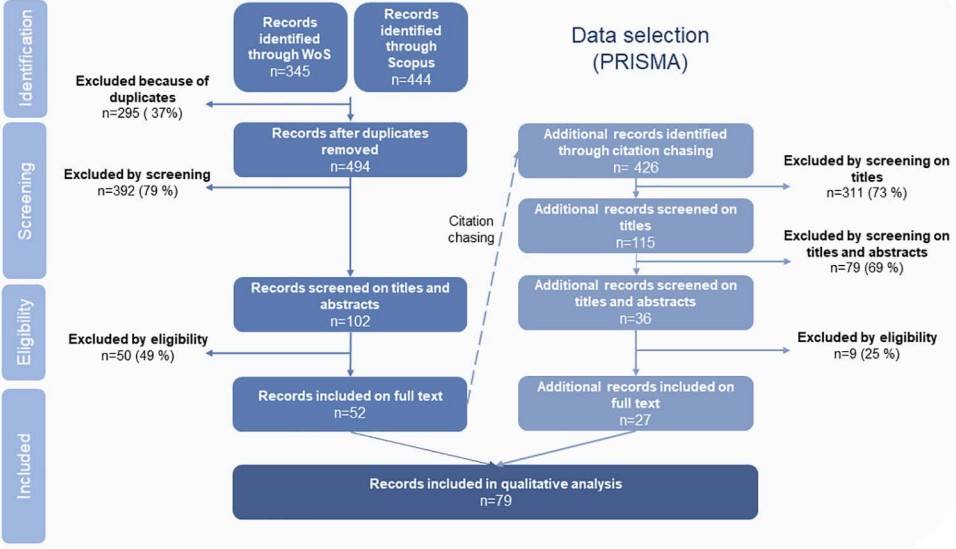

**Fig 1. PRISMA (Preferred Reporting Items for Systematic Reviews and Meta-Analysis) diagram showing our approach and the number of publications selected.** Reasons for exclusion are detailed in S2 Table.

## 2.4. Data extraction and analysis process

The elements of the results sought, summarized in the PICO table (Table 2), were collected from the papers in the corpus by the first author, following the five prisms of analysis of agricultural work in SFSCs (§2.1). Our results section is therefore divided into five sub-sections, which qualitatively summarize the results collected in the articles. Particular attention has been paid to the study contexts and methodologies. A bibliometric sub-section was also compiled by collecting the main metadata of the documents (S4 Table).

## 2.5. Risks of bias analysis

Two major risks of bias have been identified in our review. As detailed in our protocol (S1), the first bias relates to our data collection and analysis methodology. Our query may exclude certain papers of interest that do not fit our thesaurus (sampling bias). However, a "citation chasing" step was carried out to reduce this bias (§2.3). In addition, only cases that have been the subject of a study are taken into account, leaving out all the other situations that have not been studied, which are thus identified as gaps in our knowledge (selection bias). Finally, the exercise of qualitative synthesis implies a certain bias linked to the authors themselves, which tends to be limited by the establishment of a protocol and an analysis grid prior to the start of the review, as well as the use of double-blind evaluation tools and cross-analysis of discrepancies in the selection of papers (S1 Document).

The second risk of bias relates to the biases of the papers themselves. To our knowledge, there is no institutional bias analysis grid in the social sciences similar to those existing for the medical sciences [36,37]. Following Martin et al (2022) [38] and Petticrew and Roberts (2006) [30], we propose five criteria related to the method on which the publications are based, which we consider as important to characterize the SFSCs work situations studied. For each article, we will assess the presence of: elements characterizing the study area (C1); elements characterizing the farms concerned from a technical-economic point of view (C2); elements characterizing the various SFSCs employed by the farms (C3); elements characterizing the workforce involved in SFSC-related work (C4); and 5) elements justifying the sampling methods (C5). The four authors each independently assessed a quarter of the corpus.

# 3. Description of the corpus

## 3.1. The theme of "Work of farmers" in SFSC literature

Work is clearly not at the heart of publications on SFSCs. Of the 1923 articles on SFSCs in WoS and the 2363 articles on SFSCs in Scopus (resulting from a query using only the SFSC thesaurus tested according to the parameters described in Table 1, excepted for date, limited to 31/12/2023), only 17% also correspond to the "work" and "agriculture" thesauri (Fig 2).

The 79 publications we consider report on, for the most part, research conducted recently: more than 80% were published after 2016, which is consistent with the dynamics of publications around SFSCs alone (Fig 3). The relative percentage of work-related SFSC publications remains roughly constant over time.

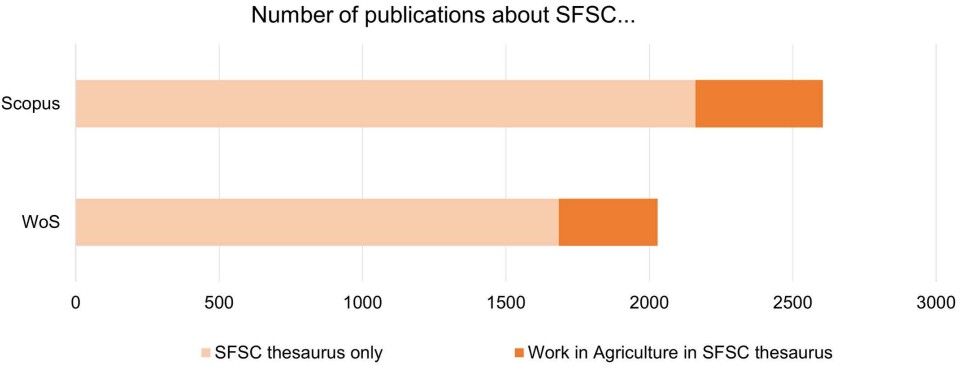

**Fig 2. Share of publications about SFSCs including the "Work" and "Agriculture" thesauri.**

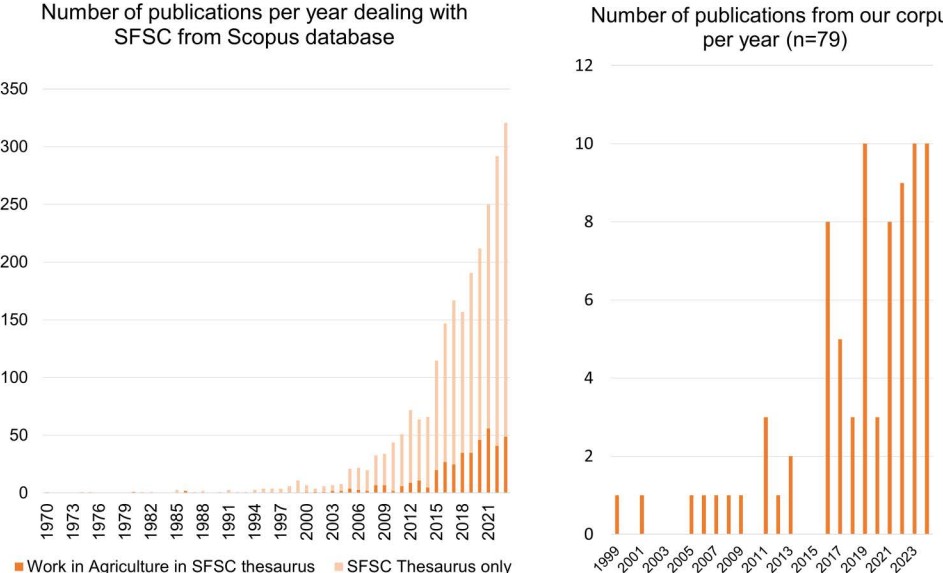

**Fig 3. Left: evolution of the number of SFSC-related publications in the Scopus database, and of the proportion including the "work" and "agriculture" thesauri in their abstract titles and keywords.** Right: number of publications per year in our final corpus of 79 articles.

## 3.2. Authors and case studies

More than three-quarters of the corpus comes from European and North American authors. 128 European authors (51% of all authors) contributed to writing 42% of the corpus (taking into account the relative weight of each co-author), and 67 North American authors (27% of all authors) contributed to 40% of the corpus (Fig 4). Other continents are less represented, with 30 Asian authors, 20 South American authors, 4 Oceanic authors and 1 African author, collectively contributing to 18% of the corpus (Fig 4).

Some countries, such as the USA or Canada, are heavily involved in studying SFSCs in general, and are similarly invested in examining work of farmers in SFSCs (Fig 4). Conversely, other countries that are major players in SFSCs research invest relatively little in question of working in SFSCs (for example Italy, China, Indonesia).

More than a third of our corpus pertains to SFSC-related work situations in the United States and Canada, with 28% of the articles focusing on Europe (Fig 5). Only seven publications study situations in South America (Brazil, Argentina, Colombia, Peru), six in Asia (Japan, Vietnam, China), and one in Africa (Senegal) (Fig 5). More than a third of the articles therefore concern Western countries, even though marketing through SFSCs takes place in all countries of the world [2]. In some developing countries, SFSCs are the main channels for marketing agricultural production [39]. The study of agricultural work in SFSCs, with SFSCs designated as such, is mostly carried out by Western researchers focusing on Western contexts.

The subject of work in short supply chains is not a topic of advanced specialization, nor does a structured research community exist around it. Indeed, these publications have been authored by 250 different researchers, only 15 (6%) of whom have published more than once on the topic, with a maximum of 5 publications for a single author (P. Mundler). The network of co-authors is highly fragmented (Fig 6). It is only because of a few recurring authors, such as economists P. Mundler (Canada), G. Feola (Netherlands) and I. Fertö (Hungary), that a few co-authorship groups have formed.

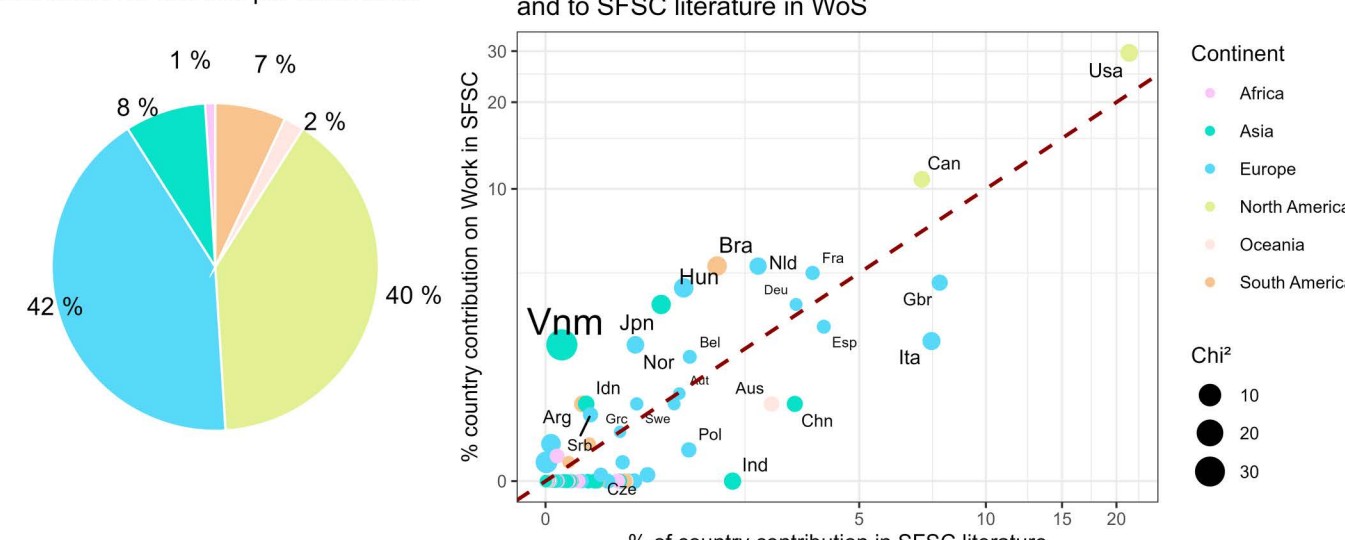

**Fig 4. Left: percentage of authors' contribution to our corpus per continent.** Right: Comparison of the countries of origin of authors working on SFSCs in Web of Science with those studying work in SFSCs (our reference corpus).

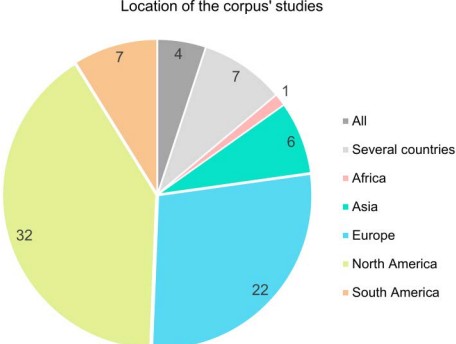

**Fig 5. Number of publications by country studied.**

Economists are the main contributors, followed by sociologists and geographers (Fig 7). With increasing number of publications on the subject (particularly biotechnical disciplines), we also note a disciplinary diversification of its study, where the part of economics seems to be holding steady, and that of sociology and geography declining relatively (Fig 7).

This disciplinary profile is also found in the main journals publishing on the topic. These journals promote multidisciplinary work on agriculture or rurality at the interface of economics, sociology and geography or in agroecology (Fig 8).

An examination of the main references cited in the corpus also shows the recurrence of studies in the disciplinary fields already mentioned (economics, sociology and rural geography) (S3 Table). Except in a few publications included in the corpus, we note the large number of references to seminal studies on AFN and the challenges of agricultural relocalization.

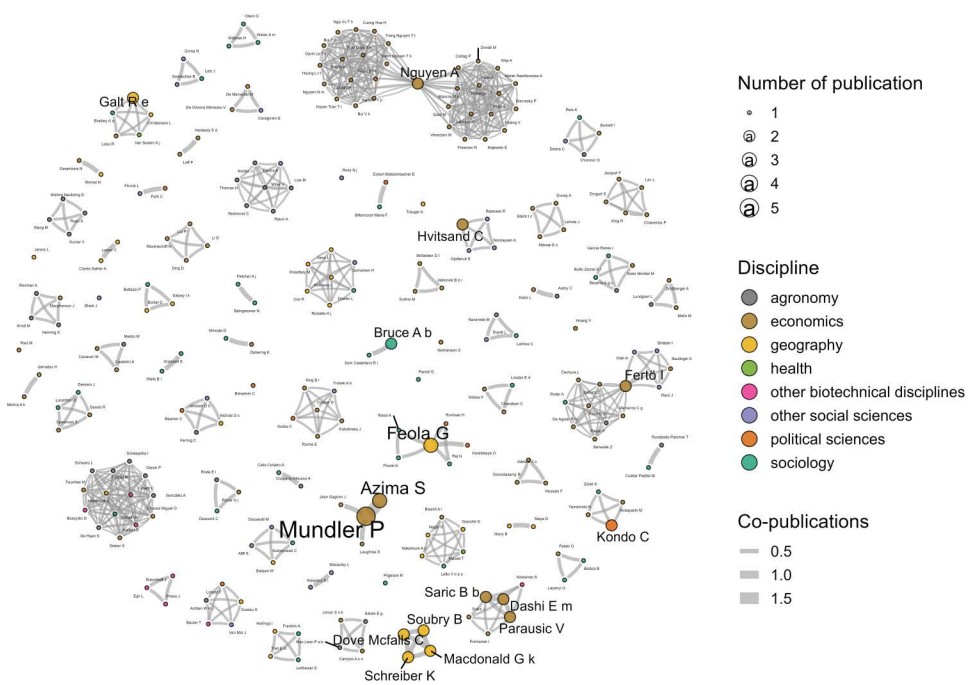

**Fig 6. Co-authorship network (graph generated using the force-directed Fruchterman-Reingold algorithm).**

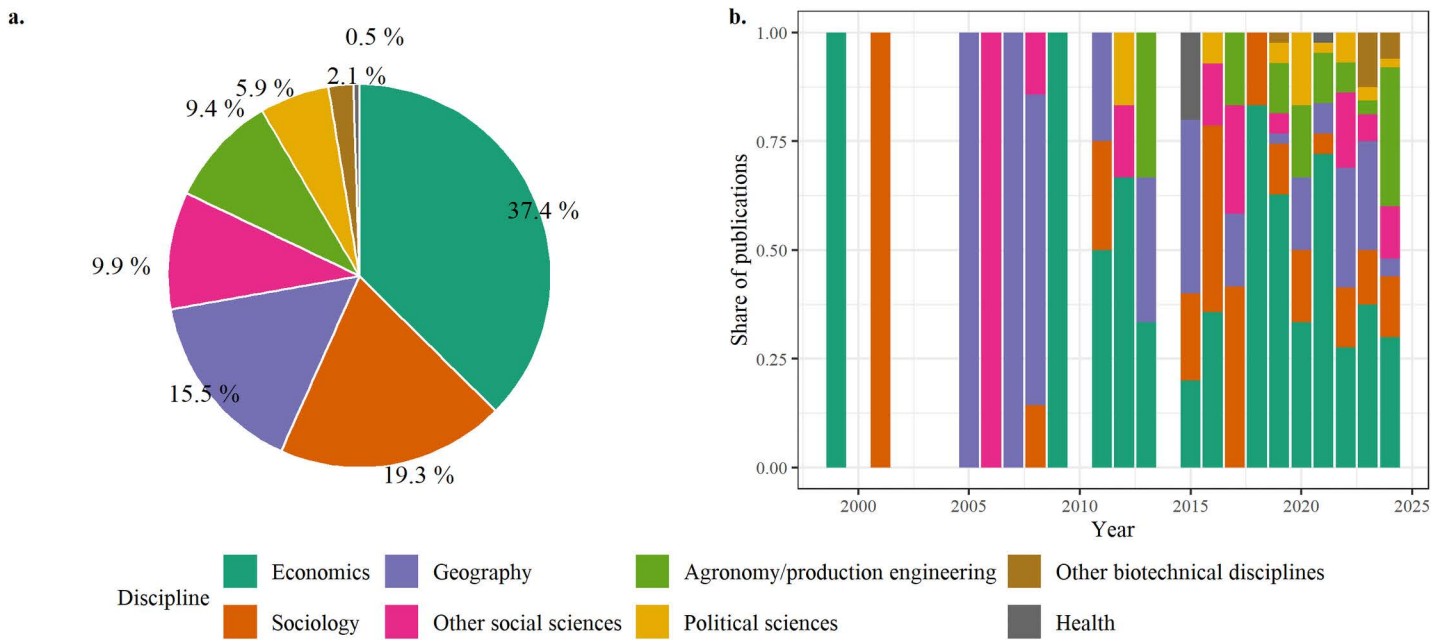

**Fig 7. Disciplinary composition of the corpus (a), and proportion of each discipline in the corpus as a function of time (b) (data on disciplines collected from the authors' self-declarations on their CVs or personal websites).**

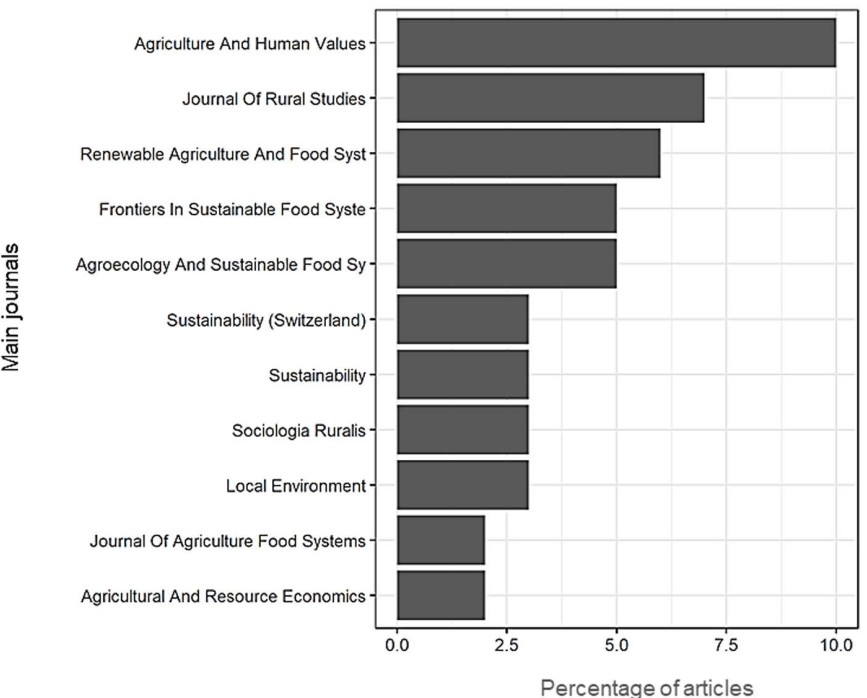

**Fig 8. Main journals publishing articles on work in SFSCs.**

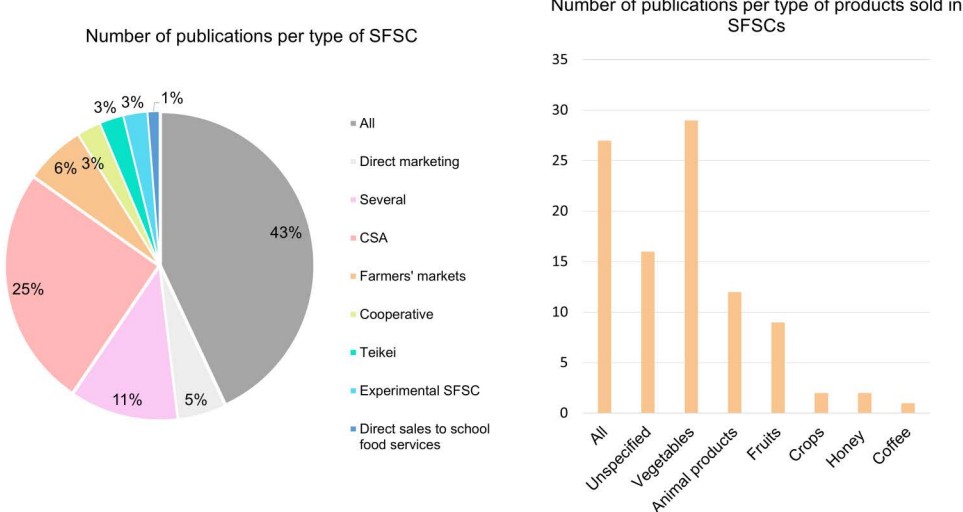

**Fig 9. On the left, number of publications in the corpus by type of marketing channel.** On the right, number of publications by type of product sold (a publication may focus on several products).

## 3.3. Methodologies and studied work dimensions

74 of the 79 articles selected for our corpus are based on empirical findings, with the remaining 5 based on literature reviews, associated with the production of a conceptual framework

for one of them. The approaches adopted in these 74 empirical articles are predominantly qualitative (53% of the corpus), based on semi-directive interviews and sometimes on participant observation in ethnographic approaches. Thirteen percent of the articles are exclusively quantitative, and 28% combine quantitative approaches with a few qualitative interviews.

The preferred scales of analysis are workers and farms, rather than territories and agricultural models. The occupation dimension is much less discussed than that of work organization and working conditions, the two dimensions most present, or those of economic performance and farm structure. Finally, 38% of the publications in the corpus have work as only one of several result elements.

### 3.4. SFSC-related work situations studied

Nearly two thirds of the publications focus on farms marketing via a variety of SFSCs, sometimes in combination with long supply chains (Fig 9). Among all the various types of SFSCs, CSAs – partnership systems between farmers and consumers that allow the risks and rewards of farming to be shared, receive particular attention: 25% of articles are devoted exclusively to them. Only 16 articles take a comparative approach between long supply chains and SFSCs. While marketing situations are often studied at a given point in time, a few articles offer a dynamic perspective [40–53].

As for the products marketed through SFSCs, 37% of the articles focus on the sale of vegetables (Fig 9). Conversely, only 15% of the articles deal specifically with the sale of animal products. Finally, one-third of the articles deal with all products traded in a given territory. The work situations addressed by the articles in our corpus are thus very varied.

### 3.5. Assessment of the risk of bias

To assess the risk of bias in our corpus, we proposed five criteria (§2.5). The detailed analysis of the risk of bias is provided in S4 Table. 52% of the articles provide clear information on 4 or more out of the 5 criteria: work situations studied in these articles are well characterized. 43% of the articles meet only 2 or 3 of the 5 criteria, resulting in a lack of precision about the studied work situation. Four articles meet only one or none of the criteria, with insufficiently characterized work situations. The territorial contexts of the studies (C1) and sampling methods (C5) are well characterized in over 70% of the studies. Only 65% of the articles include clear technical and economical elements regarding the SFSCs farms considered (C2). Less than two thirds of the articles include elements characterizing the type of workers involved in SFSCs (C4), and the forms of SFSC studied and their role among other food supply chains used by the farms (C3). In contrast to other systematic reviews on work [38], the corpus demonstrates a relatively robust characterization of the work situations studied. However, there remains potential for enrichment.

## 4. Structural elements of SFSC-related work

Analysis of the structural elements of work is not at the core of any of the publications in the corpus. Some articles use mainly qualitative approaches to examine the various profiles of workers involved in SFSCs [50,54,55], sometimes through the prism of gender [56,57]. Others report on the diversity of farms marketing via SFSCs, either in a territory through qualitative or mixed methods [58,59] or on a national scale using statistical data [60]. On a territorial scale, some also examine the determinants of the development of farms in SFSCs [58], while others focus on the various forms of employment observed [61–63]. The samples mobilized by the articles shed light on the structural forms of SFSC-related work currently documented in the literature.

## 4.1. Worker scale

Several categories of workers are found on farms marketing via SFSCs. The few articles concerning countries of the Global South refer to peri-urban family farmers on small plots of land [45,64–69]. Most farmers are involved in collective marketing initiatives, sometimes structured as cooperatives [45,53,64–67,70].

In countries of the Global North, farmers in SFSCs are reported as being more educated than the average farmer in their regions [56,71–74], with a high level of education being positively correlated with SFSC adoption in the USA [60]. Farmers from non-agricultural backgrounds, much studied by the research community [40,54,56,73–76], are also very present in SFSCs [75]. They possess a wide range of skills [54,60], and have the ability to mobilize several sources of capital for training and embarking on farming activities [54,76,77]. While SFSCs allow farmers to start off gradually with limited capital [54,72], the children of farmers returning to the land after other professional experiences are more fortunate [60,76], benefiting both from the financial and educational resources acquired in their previous professional activities, as well as from the social and financial capital of their farmer parents [54]. In these countries, apprentices and trainees, often young, well-educated and financially well-off, come to work and train on certain farms [55,62,63,78]. Highly skilled temporary migrant workers can also be found [62,78]. Finally, volunteer workers, often from privileged backgrounds [62,79,80], are also often found on certain farms, especially urban ones [77,79,81], in teikei in Japan [46,50] and in CSAs [80,82]. In the Global North as in the Global South, women are very much present [12,83], not only as farmers [42,44,67] but also as volunteers [51,59].

## 4.2. Farm scale

In countries of the Global North, farms marketing through SFSCs are very diverse They vary widely in size, ranging from one-person undertakings of less than 2 hectares [57] to farms of several hundred hectares with several dozen workers [51]. They are smaller than other farms in the areas studied [68,71,72]. Indeed, there is a positive correlation in the USA between small farm size and the introduction of direct sales [60]. Less diversified [59] and more specialized in "staple" products, larger farms benefit more from government subsidies to supplement low selling prices in long supply chains, making the switch to SFSCs less necessary [60] than for small farms [54]. In our corpus, the SFSC farms studied are also more diversified in terms of products [55,59,72,84], often offering several products [41,63,85,86].

Urban and peri-urban farms are especially well studied, both in the Global North [51,77,79,87] and in the Global South [45,64,65,69]. Within the same territory, farms in SFSCs are not, on average, specialized in the same products as farms relying on long supply chains [58,72]. Market-garden farms use SFSCs in particular [60,72] and are thus very present in the corpus (Fig 9). Animal products are less present in SFSCs in the USA [60] since they require expensive means of transport and storage to ensure product safety [64].

Many of the SFSC farms discussed in the publications in the corpus have more environmentally friendly practices than those using long supply chains [12,49,65,71,81,84–86,88–92]. In fact, some of these farms' products are difficult to market via long supply chains, such as animal carcasses derived from autonomous-economical feeding practices which often do not meet the requirements of long supply chains [86], or organic products, for which prices from long supply chains are too low to remunerate the extra work required [71]. Some marketing channels, such as CSAs and teikeis, are also closely associated with organic production methods. Finally, the implementation of an agroecological transition by farmers is frequently part of a systemic approach that they extend to marketing methods [65].

At the time of setting up in agriculture, as also thereafter, non-agricultural activities are ubiquitous on SFSC farms, both on the farm and off the farm [72,77,79,88,93–95]. In Canada, Azima and Mundler (2022) found, for example, that 75% of the 613 farms using short supply chains that they surveyed had external sources of income, and that 52% of them were involved in agritourism activities. The spouse's off-farm work often contributes to the farm household's economic stability [88,90]. Setting up in agriculture also often relies on off-farm income and capital, usually from previous employment or family inheritance [54].

SFSC farms rely on distribution through one [45,96] or several types of SFSCs [12,63,73,83,86]. Long marketing channels and SFSCs are used jointly on some farms [12,97], especially on larger farms that have more volumes to sell [83]. The case of CSA farms is well documented [51,52,55,57,69,74,79,80,82,96,98–102]. Among the diversity of sales channels, some seem to be preferred over others, such as farm-gate or roadside sales, farmers' markets and sales to grocery stores and restaurants in the USA [60], or farmer cooperatives in South America [64,65], with differences between countries. Farm size [57,59,84,96] and diversity of production also play a role in the choice of sales channels [43]. For example, CSA farms are generally smaller, with a more diverse range of products than farms using other types of SFSCs [41,96]. Available resources and skilled labor, as well as other subjective factors also have an influence [59,84].

### 4.3. Territorial scale

**4.3.1. Territorial factors influencing SFSC-related work.** The geographical proximity to consumers, determined in particular by proximity to communication routes [68,81], topography [86] and urban or peri-urban location influence the presence and distribution of farms using SFSCs [45,56,58,79,86,103]. While direct proximity to consumers favors the establishment of SFSCs, farms may find it difficult to find land or skilled labor in peri-urban contexts [58,85]. Farmers further away from consumers sometimes cover long distances to sell in larger towns and may face difficulties in recruiting labor [92].

As a consequence of historical and geographical trends [58,86], the dynamics of territorial development are also mentioned. Factors that favor SFSCs are collective slaughtering and processing facilities [60,86], the presence of intermediaries linked to SFSCs [68,86], and the development of local self-help dynamics between producers [58,70]. In particular, livestock farmers are highly dependent on intermediaries for certain stages in the processing of some of their products sold in SFSCs, given the strong regulatory and health constraints on animal products [92]. Territorial public policies [56,58,65,68,77,86,104] and national public policies [105,106,107], which steer regional development and regulate markets, play an important role. For example, some farmers question public import policies or the presence of local wholesalers, which generate competition [50,58,65].

**4.3.2. Employment.** SFSCs create permanent and seasonal jobs on SFSC farms [58,61,68,72] whether for production, processing and sales [61] and in territories in which these farms are located [60,108]. For example, CSAs have shown long-term revenue visibility and have been identified as a means of sustaining jobs [90,104]. SFSC intermediaries also find it necessary to hire workers [61]. The terms and conditions of remuneration and the permanence of these jobs are highly variable and are the subject of debate [60,82,94,105].

SFSCs also generate low-paid and unpaid forms of work [62,63,77,79,82,87,88]. Low sales prices limit hiring opportunities [78]. There are marked differences between the type of productions: dairy farms tend to eliminate precarious labor once they become profitable, while vegetable farms depend on precarious employees throughout their life cycle for their profitability [94]. Even though apprentices and trainees come for training and to gain experience, they are perceived primarily as cheap labor [63], much like seasonal migrants, who

are recognized for their skills but do not enjoy the same rights as domestic workers [62,78]. Finally, some urban farms and some CSAs rely structurally on the work of a large number of volunteers [77,79,82,87].

## 5. Economic performance of work

Various methods have been used to determine economic performance. We refer to two literature reviews on CSAs [107,109], one meta-analysis [110], quantitative surveys of a small [64,90] and a large number of farmers using SFSCs [72,83], and to articles using qualitative economic satisfaction indicators [71,75,88,93]. The performance of work has been addressed via the amount of work devoted to SFSCs [71,89,96], value added or revenue generated [60,72,83,89–91]. Some articles adopt an approach of comparison between short and long supply chains [68,72,83,91], while others examine the viability of farms [12,64,90].

### 5.1.  Working hours and hourly pay

5.1.1.  **Working hours.**  On a qualitative level, farmers using SFSCs say they work a very large number of hours per week [71,72,76,94], with sometimes long working days [76], and lament an inability to take vacations or even weekends off [71]. On a quantitative level, working time required for sales activities is higher than for long marketing channels, especially because of the additional time needed for packaging, transport and sales [83]. Working time varies widely between farms, depending on the chosen sales channels [83,96]. Production activities also require a sometimes very high hourly volume [89], all the more so when farmers in SFSCs are engaged in alternative production practices (§4.2). Finally, the "time left over" for other activities after on-call work varies greatly according to the organizational choices made [89]. Since the time allocated to production is often not very compressible and relatively inflexible, it is from marketing activities that farmers in SFSCs tend to try to shave time [89]. None of the articles examines the time required for processing of products destined to be sold via SFSCs.

5.1.2.  **Hourly labor productivity in SFSCs.**  Even though they vary from one channel to another, SFSC price levels are always higher than in long chains [70,71,83,91,111]. In direct marketing channels, prices are often set by the farmers themselves, and are mainly based on observed market prices, rather than on production costs [94]. Indeed, prices are limited by consumer purchasing power [76] and by competitive effects between farmers [95,100,102]. They are sometimes discussed between farmers at collective selling locations [112]. Some CSAs and teikeis, based on principles of solidarity with farmers, take account of the farmers' labor costs in their prices [82,87,95,100]. In SFSCs with intermediaries, the intermediaries retain considerable pricing power [94,108]. However, marketing costs are often higher in SFSCs [91]. Most notably, they are dependent on the cost of salaried labor and transport[70,91]. Prices vary from one sales channel to another, and are, for example, higher in farmers' markets than in CSAs [91,96]. Packaging and sorting standards, product diversity to satisfy the customer base, marketing approach and benefits extended to good customers also play a part in these costs [91]. In six European countries, SFSCs offer higher added value than long marketing channels, mainly due to higher prices levels [83].

Labor productivity (value added/hour worked) therefore varies widely and depends on sales strategies, production choices and work organization [89]. The processing of a wide range of products, including fruits and vegetables [89], and sales via farmers' markets rather than CSAs [96] have a negative impact on labor productivity. The farms with the highest net incomes are those with the highest labor productivity in processing and marketing activities, the latter compensating for generally lower labor productivity in production activities [89].

### 5.1.3. Hourly pay.

While no article in our corpus calculates a quantitative indicator of hourly income, many farmers using SFSCs are dissatisfied with it and consider it too low [88]. Some researchers note that SFSC prices are insufficient to adequately remunerate the extra work generated by the use of SFSCs [68,71]. The objective of making products accessible to as many people as possible often has a detrimental impact on remuneration for working time [82]. When labor costs are taken into account when in calculating prices, the legal minimum wage paid to unskilled workers is used in the calculations, and the complexity of the tasks and managerial responsibilities carried out by farmers and their employees is not taken into account [82]. Selling via CSAs [96,109], coming together in cooperatives [68], helping each other out, lending equipment and using self-produced inputs [64,65,68,86,87] can, however, reduce costs and improve hourly remuneration. Hourly income does not systematically correlate with the overall income of salaried employees [70,90] since they may receive other non-monetary benefits.

## 5.2. Income

### 5.2.1. Income levels.

The results also diverge when it comes to income uncorrelated with working time. For example, some CSA farmers report higher income levels than if they marketed via other channels [55,99,100] while other farmers feel that CSAs do not remunerate them enough [82,95,102]. Among the articles studied by Egli et al. (2023), more than half present inconclusive results, while the rest are split between positive or "negative or no" conclusions regarding increases in income due to SFSCs. These divergences are first and foremost due to methodological choices. Therefore, qualitative methods for the assessments of income tend to give more positive results than quantitative ones [110].

These differences are also geographical [107,109]. For example, European publications show economic performance to be superior to those from the US [110]. Finally, for the Global South, several articles suggest that farmers have increased their incomes significantly by implementing SFSCs [66,68,70]. Differences in farming systems and agricultural policies between countries, as well as market conditions are cited to explain these variations [50,64,74,90,110].

There are very large income disparities within the same geographical areas [72,90]. The effect of the type of production appears to be significant [44,72,89]. In a Quebec field study, dairy farms using SFSCs generated higher incomes than those producing meat or fruit and vegetables [89], but only market gardeners seem to earn a higher income by using SFSCs as compared to using long supply chains [72]. On the other hand, no form of SFSC sale is associated with clear positive impacts on income [70,89]. The same is true for the adoption of multiple forms of marketing [60]. Income disparities seem to be present in all modes of sales through SFSCs [110]. Finally, farm size is positively correlated with income, possibly due to economies of scale [44,72,90].

### 5.2.2. Income stability.

If SFSCs don't necessarily increase income, they can help stabilize and secure it [12,52,55,66,99,100,108] by freeing producers from a certain dependence on intermediaries and the volatility of world prices [52,66,70]. Thanks to regular consumer contribution in CSAs, producers have a greater visibility of their revenues [81,95,100,104] and can invest while limiting their borrowings [81,104]. Indeed, this solidarity is the basis of the CSA concept [81,95,100,104]. However, consumer involvement varies from one CSA to another, and is tending to decline: consumer expectations are changing [46], and producers are making the conditions for consumer participation more flexible, while competition between marketing channels is increasing [95,102]. In intermediated SFSCs, the longer the intermediaries are involved with producers, the lower is the purchase price [94]. In the end, farmers are still largely responsible for managing vulnerability.

During Covid, farms in SFSCs were exhibited some economic resilience [44,45], although it was varied between farms and countries [44,53]. However, farmers have largely had to reorganize their work in the face of growing demand and the closure of certain sales outlets [48,53]. Additional costs have not fully been passed on to consumers [40,53]. Sales via SFSCs finally remain sensitive to any fluctuation in the local market [50,64,76,91,102], with difficulties particularly acute in developing countries [53].

**5.2.3. Indirect indicators of income levels.** While some farms in SFSCs enjoy high levels of income, a significant proportion of the farmers studied have very low incomes [56,90,104]. Galt (2013) introduces the term "super self-exploitation" to designate cases in which the farmer's income is lower than that of his employee [90].

Farm households adopt various strategies to secure their incomes and limit their expenditure. In the Global South, for example, some farmers using SFSCs resort to self-consumption and non-market exchanges [53,64], and rely almost exclusively on family labor (§4.1), whose remuneration can be more flexible [45,64–69]. In countries of the Global North, while many farms using SFSCs rely on family labor [54,113], there is frequent recourse to low-paid or unpaid forms of work, which is sometimes even structurally essential [56,62,63,71,78–80,82,92,94,95,105]. Part of the work is, thus, underpaid. Finally, some farmers derive income from other activities [12,54,75–77,81,82,94,95]. Some also benefit from the income of their spouses working outside the farm [12,75,104] or from capital accumulated before embarking on agriculture or received through inheritance [54,76,95]. The articles in our corpus do not specify whether these strategies are more widespread among farmers using SFSCs than the average for all farmers. Are they the result of a deliberate choice on the part of farmers using SFSCs in search of other ways of farming, or are they symptomatic of a model that struggles to adequately remunerate work? Some studies [54,76,90] call these models' economic performance into question, which are less supported by governments than conventional models [86].

## 6. Work organization

The articles in our corpus approach the organization of work from two angles: that of tasks and their articulation over time, and that of the social relations of production between workers. This theme has been explored in a literature review [107] and in territorial case studies focusing on specific supply chains [50,64,65,74,90] or specific productions [58,85].

### 6.1. Work rhythms and organization of tasks

**6.1.1. Tasks and skills associated with marketing via SFSCs.** Marketing via SFSCs involves specific marketing and processing tasks [68,91,96,108] but none of the articles dwells on their spatial and temporal organization. Added to the usual production activities, these tasks are particularly time-consuming [72,91,93] and their durations are difficult to predict [89]. Furthermore, they require specific skills. For example, building customer loyalty [81] requires the ability to assess customers' expectations [50] in order to implement a marketing approach [59,74,81] and an organization suitable for the chosen sales location [74,107]. Product packaging and labeling have to conform to regulatory and health standards [59,68,91]. Often managed by the farmers themselves [97], logistics activities are also time-consuming. Farmers are often unable to anticipate them fully [71] since there are many parameters to consider, most notably physical distance to consumers [91,96,103] and to slaughter/cutting intermediaries for animal products [59,86,97], the volumes to be transported and health regulations [59]. For dairy and meat products, certain processing stages are delegated to intermediaries in the supply chain, which means establishing working relationships with a variety of stakeholders [97]. Processing is hardly mentioned in the

publications in the corpus. Since each SFSCs outlet has a limited absorption capacity [83], many farmers using SFSCs combine several types of marketing channels (§4.2). Consequently, the work increases and its organization becomes more complex [59], even if this strategy makes it possible to take advantage of complementarities between products to facilitate their sale [58,59,74] and disperse risks [97].

Finally, the organization of SFSC-related work is highly dependent on the weather and changes in socio-economic context [50,64,76,91,102], with difficulties exacerbated in countries of the Global South [53]. During the Covid period, for example, the closure of certain sales outlets, increased demand for local produce and difficulties in accessing seasonal labor forced producers to modify their organization of work, in order to ensure food supplies, guarantee consumer and worker health safety and secure the economic equilibrium of their operations [40,47,53].

**6.1.2. Impacts on the organization of agricultural production work.** Marketing via SFSCs also impacts the organization of production tasks [59,89,106,114]. To meet demand, some farmers first increase production volumes, even if this means an almost unmanageable increase in workload [93]. Direct-sales market gardeners also sometimes increase crop diversity. This strategy represents a challenge in terms of know-how and workload [85] but allows them to spread their risks [104] and satisfy customer demand [85]. Finally, crop calendars are sometimes designed to offer a diversity of products at specific times of the year [49,85]. In the coffee sector, improving quality to be able to sell via short supply chains requires the adoption of new harvesting practices that require increased labor and additional equipment [70]. Finally, for some producers, marketing via SFSCs is tied to the adoption of more environmentally friendly production practices (§4.2), which are often more labor- and skills-intensive [65,71,95], even if some are designed to limit labor time [77].

**6.1.3. Work organization, consumer expectations and farm requirements.** If they are to reduce their dependence on long supply chains, farmers need to take account, in their work organization, of the expectations of consumers and intermediaries in SFSCs. Direct sales enable farmers to meet consumer' expectations directly [115]. These expectations in terms of volumes, product diversity and supply schedules are sometimes at odds with the way the farms operate [49]. To meet these expectations, some farmers in CSA, for example, feel obligated to supply the same quantity of produce, with the same regularity [74,90,95,107]. Some producers also cap their selling prices to remain affordable and agree to staggered payments, to the detriment of their organizational convenience and earnings [76,95,100,102]. According to Galt (2013) and Birtalan et al. (2022), constant contact with customers ultimately pushes some farmers to make organizational decisions that are detrimental to their own interests [74,90]. Their decision-making autonomy is curtailed, whereas the initial values proclaimed by AFN promised the very opposite [90]. On the other hand, some CSAs involve consumers in work and decision-making [80]. The organization of work is discussed with them in order to satisfy both consumer and producer expectations [80].

## 6.2. Coping with the workload

Faced with the heavy workload associated with marketing via SFSCs, farmers are changing not only their production, processing and marketing methods, but also their working collaborations. Territorial actors are also getting involved in the implementation of SFSCs.

**6.2.1. Reorganizations of tasks and associations.** At the farm level, agricultural and marketing work is frequently reorganized, not only to save time, but also to reduce the associated mental workload [85]. Market gardeners, for example, re-specialize their production, reorient their sales outlets, and synchronize their commercial activities with their agricultural calendars [85].

Various digital tools can be used to manage customers, advertise and rationalize work organization [12,47,48,107]. Their impact on working hours is the subject of debate [12,48]. Designed from a capitalist perspective, they influence the way farmers think about work [48] and are sometimes perceived as incompatible with SFSC values [12].

At the territorial scale, intermediaries in SFSCs take on marketing-related tasks, such as door-to-door sales and managing sales outlets [92,94]. Some producers also join forces through more or less formal collaborations, which enable them, for example, to offer a wider range of products, share infrastructure, pool journeys, even undertake processing together, or simply provide moral support to each other [49,57,59,65–67,87,92,93,103,108,115]. These associations are often built around common interests [59] and close relationships [66] – a guarantee of commitment and trust. However, group work requires skills that some producers lack [59]. Some also lack the time to enter into new collaborations and to reorganize their work accordingly [49,57,65,93,103] while others do not feel that they cooperate more in SFSCs than in long supply chains [72,88].

**6.2.2. Mobilizing the workforce.** It is primarily family labor (§4.1) that takes on the burden of the high workload associated with SFSCs. The family workforce's inherent flexibility in terms of the duration and periods of work is best suited to address the fluctuations in work schedules specific to SFSCs [85]. As this workforce sets its own prices for its labor, it is flexible in terms of remuneration, which enables it to cope with contingencies and control cash flow requirements [41,42,64]. Family labor helps to keep farms in operation, in a context in which price levels are too low to remunerate employees [71,94]. However, this situation too often leads to "self-exploitation" by farmers, denounced by Galt (2013),Bruce and Castellano (2017) and O'Connor et al. (2024) [12,71,90], who draw on the work of Tchayanov (1925) to question the sustainability of these models (§ 5.1.3). In these family groups, the division of tasks is sometimes highly gendered [64,78,80], with variations depending on the country. For example, women are heavily involved in sales, packaging products [83] and setting up collective sales outlets [66,69]. They seem to have greater autonomy than in long distribution channels [55], but are sometimes excluded from decision-making, management and technical tasks [78]. Their key role is sometimes little recognized [66].

Other categories of workers are involved in the work collectives of farms using SFSCs. Although salaried employment is mentioned, it is not the focus of any of the articles and is only analyzed in terms of employment [58,61,94,107]. Salaried employees are involved in production, processing and marketing, and sometimes in team and project coordination [77,79]. Sometimes part-time or temporary employed [77,81,94,95], their remuneration levels are more or less "fair" [61,77,81,82,94].

Receiving little or no remuneration, trainees and apprentices enjoy a certain degree of autonomy in their work organization, and are sometimes even assigned management responsibilities [62,63,77,78]. On family farms, they emulate the farmers in their activities, including family life [62,63,78]. However, they work long hours, "self-exploiting" in the same way as farm managers or owners [62,63,78].

In comparison, seasonal migrant workers, who come under the ambit of strict bi-national agreements, have far less autonomy. Technically highly skilled, they are often assigned to tasks that locals do not want to do [62,78]. They are paid, but do not receive the entire range of social benefits [62,78]. Their work relations with farm managers are highly asymmetrical, as the managers have the power to send them back to their home country [62,78].

Finally, volunteer workers can be found on urban farms [77,79,87], as consumer-members of CSA or teikei [46,50,52,80–82,95,104,105] or on family farms as "volunteers" or "wwoofers" [76,88,92]. They can be involved on an ad hoc basis (major projects, weather contingencies, exceptional situations such as Covid) [41,43,104] or on a regular basis (weekly volunteering,

long stays, etc.) [50,79,80]. They are sometimes assigned decision-making responsibilities [46,50,80]. The ways of their involvement evolve considerably over time, in line with changes in socio-economic contexts [46,95,102]. Farmers organize the work to take account the volunteers' availability, and their skills and aptitudes [50,79,80] within a framework that provides them security [81] and comfortable work conditions [80]. These volunteers are recruited through direct sales, social networks and convivial events, that are specially organized to keep the community spirit alive and attract new members [46,79–82,95]. Although mostly unpaid, volunteers enjoy a variety of benefits, including cheaper products [50,105], hands-on training, integration into a professional network, etc. [79,87].

**6.2.3. Involvement of other actors at the national and territorial levels.** Local and national non-farmer actors indirectly influence the organization of SFSC-related work. At the territorial level, local authorities allocate sites for sales points, and work alongside NGOs to promote local products [56,70]. NGOs, researchers, advisors and farmer cooperatives provide support for setting up sales outlets [49,58,64,65,67,69].In France, regional councils provide subsidies to employer groups to recruit manpower [58]. In the particular case of Covid, local intermediaries participated in product distribution [42].

Governments also support the development of SFSCs at the national and supranational levels through financial and technical incentives [107] and through institutional recognition [81]. This support is primarily aimed at facilitating access to manpower [62,87,94] or targeted equipment [68,110], or to support the development of direct sales [66,68,110]. Collective farmer organizations are sometimes targeted as a priority [65,66,108]. However, some authors debate the relevance, efficiency and effectiveness of such support. Virtually absent in some countries [113], this support is widely perceived as insignificant compared to agricultural subsidies and other support for long supply chains [54,65,106,107]. These forms of support are often ill-suited to the sometimes "non-standard" projects carried out by farmers using SFSCs [106]. These policies support the continued existence of farms to the detriment of the workers' working conditions [62,78,94]. The neoliberal economic policies of certain countries are also called into question, as they do not protect small producers from competition [50,54,64,107]. Finally, Galt (2013) and Bruce and Castellano (2017) point out that farms using SFSCs remain rooted in a global capitalist political economy [71,90]. In their view, this context is not conducive to the sustainable development of methods to market products via SFSCs since it does not place an exchange value on the extra labor associated with the implementation of production and marketing practices that are alternative to those in the dominant model (§5.2).

## 7. Working conditions

Working conditions are mainly approached through the prism of motivation and job satisfaction. They are sometimes assessed according to pre-established criteria[40,72,75,88,114,116] and sometimes discussed through qualitative interviews [50,56,57,62,66,71,73,74,76,79,85,103,112,117]. Marketing via SFSCs clearly contributes to farmers' job satisfaction [52,66,72,73,75,85,89,94,109,116]. There exist various reasons of satisfaction and they are widely discussed. Just like motivations, they depend in particular on a person's subjectivity [57], life history [50,54,77,103], age, social class, marital status, geographical context, level of education [56] and gender [56,66,76,116]. Highly present in SFSCs (§4.1), women in particular are more satisfied with SFSCs than are men [116]. Few articles examine the physical arduousness of work [72,76,116].

### 7.1. Complex and multiple tasks: mental workload, autonomy and gratification at work

SFSCs can engender physical and psychosocial risks [12]. Marketing via SFSCs requires activities that are sometimes repetitive [85] and relies on a complex work organization

(§5.1.1; §6.1), generating a heavy mental workload [88] that varies depending on context [65,73,89,93,102]. Combined with long working hours, this mental workload is a source of stress and even burnout for some [12,76,85,88,89]. While not systematically detrimental to work satisfaction [76,85,88,89], it can, however, limit work fulfillment [76]. Women, who are more often in charge of domestic duties, may find it especially difficult to combine domestic and agricultural work [66,116]. Furthermore, product processing and marketing via SFSCs can be physically demanding [76,89,94], although some farmers consider that their working conditions are beneficial to their health [55,99,108].

Nevertheless, the diversity of SFSC-related activities is perceived as stimulating, less routine, more creative [89,106] and is a source of motivation and satisfaction. Farmers find pride in being able to set up complex systems that allows them to accompany the product through all its stages [68,73,85,89]. The many new learning experiences required also contribute to satisfaction [50,79,85,88,104,115]. The lack of management and entrepreneurial skills can, however, be disadvantageous to full participation in SFSCs [118]. Despite scheduling constraints [85], farmers are also satisfied with the greater management autonomy they enjoy by choosing to market via SFSCs [72,85,88,104], even if they remain subject to consumer demands (§6.1.3) and a legislative and administrative framework over which they have no control [55].

## 7.2. Social Satisfaction

By engendering a variety of local relationships, SFSCs help to renew relations between actors in a territory, which is a major source of motivation and satisfaction. However, some actors can develop a certain dependence on these relationships, which can cause tension [12,85].

**7.2.1. Relations with colleagues and other workers.** SFSCs provide a solid foundation for mutual support between farms (§6.2.1), especially due to sales through collective outlets [112]. They contribute to peer recognition [94], to which SFSC producers are particularly sensitive [84]. However, SFSC farmers are subject to competition between themselves [12,102,115], which can lead to stress and insecurity [12,102]. Although women are a driving force behind SFSCs, they are sometimes left out of the limelight [66] or even obstructed in their participation [12,55,78,116]. In other cases, SFSCs contribute to a reduction in ethnic and gender discrimination [108].

Farmers using SFSCs also express more satisfaction when they use salaried staff, trainees or volunteers [63,87,88], but the hierarchical and economic relationships they have with these actors are not always of foremost importance to them, and are described as not very balanced [54,62,63,76,90]. This situation is legitimized by workforce exploitation, being described as a "necessary step" in achieving the goal of transforming the system [62,63,77,79], as a part of the affirmation of a certain "moral economy" (a concept derived from the work of historian E.P. Thomas) [54,62,63,76,90]. Some farmers, sometimes self-exploiting (§6.2.2), believe that they are participating in reciprocal exchanges, as they impart agricultural training to trainees and volunteers [87,92] and offer seasonal migrants higher wages than they would get in their home countries [62]. In some CSAs, however, processes have been initiated in an attempt to move away from capitalist labor relations between producers and other workers [80,82,105].

For their part, volunteers perceive SFSCs as a way of meeting people [50,80] or escaping from an unfulfilling daily salaried life [79]. SFSCs offer them an opportunity to obtain low-cost training [63,79] and to support farmers' efforts to change the dominant agricultural model [46,50,82]. However, these low- or unpaid work experiences are limited to individuals who have the time and financial means to do so [63,77,79,80].

**7.2.2. Relationships with consumers.** Being in direct contact with consumers is a major source of motivation [46,57,98,100,112,114,117]. Farmers have high expectations from their relationships with consumers [76,112], and these relationships are significant vectors of social

recognition [68,76,88,94], a pillar of well-being at work [89]. Farmers using SFSCs are proud to be able to talk about their products and practices [68], and to build trusting relationships [99]. Sometimes, they have an aim of educating people about food [48,56,57,98,100] and of promoting a positive image of farming profession [84]. Job satisfaction from using SFSCs is specifically attributed to direct sales [88], in particular via farmers' markets [117] and CSAs [52,109], and is reinforced by alternative practices [94]. However, creating a good relationship with customers requires special effort (§6.1.1). Communicating with customers who are far removed from the agricultural world and who are sometimes inconsistent in their consumption habits can lead to difficulties (stress, emotional ups and downs and a loss of self-esteem) [40,74,85,88]. Farmers using SFSCs remain economically dependent on a certain relational proximity to their customers [12,74,95], and sometimes adopt practices that are detrimental to their own working conditions to satisfy their customers (§6.1.3).

## 7.3. Economic satisfaction

On the economic front, many farmers are happy to be able to set their own prices [72,88], and sometimes to sell for more thanks to the elimination of intermediaries [66,112,117]. In some contexts, receiving a monetary income also helps in the recognition of their activity as "work", especially for women [50,64,66]. Some also participate in SFSCs in the hope of earning a better living [54,90,117].

Economic satisfaction, however, is not always a given [88,89,102]: the economic added value of SFSC compared with long circuits is open to question (§5). In Belgium, none of the SFSC sales channels studied by Sureau et al. (2019) offers an equitable sharing of value, primarily due to governance and transaction methods that are disadvantageous to farmers [94]. In Canada, as gross farm output increases, enjoyment of work decreases, with farmers being able to achieve increased output only at the cost of more stressful, physically demanding and time-consuming activities [88]. Economic instability is a source of stress, and is sometimes only tolerated in the hope that income will improve over time [106].

However, economic satisfaction is not always a primary goal for farmers participating in SFSCs [12,52,56,57,66,84,87,89,90,92,93,104,106,114,116]. That said, the majority of them want to be able to "make a living from their activity" [12,73,106] and consequently abandon unprofitable marketing channels [57,90]. The economic motivation, while important, may be secondary to other motivations [57,106]. Two explanations are put forward for these results: first, the fact that a certain number of farmers using SFSCs have other sources of income [90,106], and, second, the fact that farmers and workers participating in SFSCs are immersed in a certain rhetoric of "self-giving" that encourages self-exploitation for the sake of an ideal [77,90].

## 7.4. SFSCs as an alternative ideal

Commitment to SFSC-based farming is often seen as a life project by farmers, sometimes associated with a political project [12,50,56,64,65,73,77,82,87,93,100,103,105,106,117]. Some, especially women, prioritize a life project centered on the family, with the objective of providing it with healthy food and of having time to devote to children [50,56,57,66,93]. Others seek to free themselves from work in other professional spheres, which they perceive as alienating [56,76,79,87], and highlight SFSC activities, conducted outdoors in connection with nature and with the company of other like-minded people [50,56,57,79,92]. What is sought in work is no longer solely economic, but also axiological [56,57,66,79,87,89,90,92,93,100,104,106,116]. Some farmers see participation in SFSCs as a way of escaping from an imposed and stressful work rhythm in long supply chains [56,57,79,93,106] and being more in control of their own

schedules [106], reducing their working hours [93] and enjoying a better work-life balance [116]. Certain sales mechanisms, such as CSAs, are being used to experiment with other forms of work relations (hierarchy, relationship to monetization of work, place of reproductive work, etc.) [12,80,82,87] and are sometimes perceived as improving the quality of working life for producers and their employees [52].

Other farmers, especially women [70,81,103,104] and individuals from outside the agricultural world [54,63,87,106] are also getting involved in SFSCs in a quest for a different agricultural model and a new food system. In opposition to conventional globalized agriculture, this system is based primarily on a more sustainable use of resources [57,73,100,116]. Its aim is to feed the local population with local and healthy products [57,76,84,98,100,103,104,114,115] and to make them accessible to as many people as possible [40,82,102]. Emancipation from the intermediaries of long supply chains should lead to fairer sharing of value and greater autonomy [66,85,92,93,100,103,116]. Some are also politically militant [50,64,65,100], or train new farmers [77,87,104] to extend an alternative agricultural model. This model's adoption, however, entails working conditions not always beneficial to farmers and to other workers involved in SFSCs. Even if many farmers consciously sacrifice their work comfort for the sake of feeling useful [40,76,87,90,92,106], some also claim their right to make a decent living from their work [66,73,104,106]. These latter producers feel a lack of support from consumers who are far removed from the realities of the farming world [76,94,107,117], and are increasingly less involved as full actors in CSAs and teikeis [46,95,102]. Intermediaries are also singled out as not willing to participate in a fairer sharing of value [94]. Public policies are perceived as unhelpful since they do not generally favor alternative agricultural models [55,117]. Finally, some authors question the very sustainability of SFSCs and the current capacity of farmers using SFSCs to free themselves definitively from the dominant capitalist system [90].

## 8. Occupation

Little studied, the "occupations" theme is approached exclusively through the prism of knowledge to be acquired and skills to be learned. When they participate in SFSCs, farmers are no longer just producers, but also technical salespeople and work organizers [65]. They come to understand that it is essential for them to master a wide range of knowledge and skills: knowledge of cultivation practices; ability to learn and innovate, to think in terms of systems [106]; to network [106,118]; proficiency in communication and marketing [113,118]; and business management [118]. Farmers, especially older ones, those on smaller farms, or those who come infrequently in contact with customers [113] often lack management skills [106], as also communication and marketing skills. Mastery of digital tools such as online sales platforms, spreadsheets and other management or marketing software is becoming increasing indispensable [47,48,106,113].

Learning is time-consuming, and is unevenly accessible [59,70,106]. Farmers' children learn first as they grow up and help out on the farm [54,64]. Before their setting up in agriculture, future farmers also learn through voluntary work, internships or apprenticeships [56,62,63,76,79]. Urban farms in particular are very much present in these mechanisms, acting as project incubators and generators of alternative networks [63,77,79,87]. Since they offer little or no remuneration, these internships and voluntary positions are accessible only to people with means [63,76]. Many apprentices, even though selected on the basis of their prior skills, lack the financial resources to start farming on their own [63]. And, even after farmers manage to embark on their own farming journeys, they find that many skills are acquired only experientially [56,85,106,115] and that some learning is difficult to objectify and pass on [85]. Farmers perceive public and private extension services as too theoretical or ill-suited to farms using SFSCs [106], and so ignore them [85,106]. They prefer instead peer exchanges on social

networks or within discussion groups [85,106]. Producer organizations (cooperatives, collective sales outlets, labels) also organize training courses and are arenas for sharing information [65,66,70,92]. Finally, Charatasari et al. (2019) call for the creation of new spaces to help farmers train to develop all the various skills required for SFSCs [118].

## 9. Discussion and research agenda

### 9.1. Transversal analysis

Work so far represents a minor theme in the study of SFSCs. However, with the internationally research community taking a growing interest in the topic of work in agri-food value chains [119], we can only hope to see an increase in the number of publications on this subject.

Our corpus covers the diversity of work configurations in SFSCs around the world in a skewed manner. The majority of authors interested in SFSC-related work are based in North America or Europe, and study these geographical regions much more extensively. The available knowledge in English on cases in Africa, Oceania, Asia and Latin America therefore needs bolstering. This disparity is also due to a more pronounced focus on market gardening, which is also very more present in SFSCs, and on sales via CSA, which is not the form of sale most frequently used by farmers in SFSC [60,120]. Cases involving different products and sales channels need to be documented to better reflect the existing diversity of work situations prevalent in SFSCs.

From a methodological standpoint, a greater diversity of approaches is certainly desirable. More quantitative studies will complement the findings of qualitative studies, since the results obtained may differ depending on the approach used [110]. Similarly, diachronic studies will provide a better account of the existing lability of organization of SFSC-related work over the long term [121,122]. More studies conducted on a territorial scale will be useful in gaining a better understanding of the specific nature of SFSC-related agricultural work in inter-farm collaborations and those between farms and other actors. The study of public policies impacting the work of farmers in SFSCs would also merit further investigation. Similarly, more studies at the scale of agricultural models present in SFSCs will lead to a better understanding of their diversity and the different work configurations associated with them. There is a need to document the occupational dimension in greater detail in order to examine the possible emergence of new occupational identities due to participation in SFSCs [123]. Finally, the assessment of the risk of bias also indicates the necessity of enhancing the characterization of the work situations studied (§3.5).

### 9.2. Structural elements of SFSC-related work

None of the articles in the corpus focuses on the structural elements of SFSC-related work. The information available on this theme varies between different geographical areas, which limits the ability to draw general inferences. However, there are some constancies between the different countries of the Global North that have been studied.

While the term SFSC encompasses a wide variety of work situations, little information is available on the different technical-economic categories of farms in SFSCs in agricultural territories, and on what differentiates them from farms in long supply chains – with the exception of one study carried out in the USA [60] and of studies in other languages [124]. Studies that take into account the structural diversity of farms in SFSCs will be essential to a better understanding of the differences in economic performance observed within the same territories, or the diversity of work organization. In this respect, for example, very little research has been carried out on salaried workers, even though they are well represented in SFSCs [60]. Their role needs to be better documented, as the proportion of salaried workers is growing in a number of countries [125] and is becoming a labor issue in its own right.

While several territorial factors are noted as having an impact on the work of farmers using SFSCs, none of the publications focuses on these factors' influence on work. Research in this area could be of interest, in particular to inform territorial development policies.

### 9.3. Economic performance

Only a small number of quantitative studies have calculated the economic performance of farms that use SFSCs. Since qualitative studies tend to show more positive results than quantitative ones [110], more quantitative approaches are needed to gain a better understanding of this metric. In particular, the issue of working time devoted to SFSCs, and hence hourly work performance, deserves better exploration from a quantitative point of view. It is a major factor for job satisfaction and long-term farm viability. Product processing stages, which are often necessary for animal and crop products, should also be included in studies on the economic performance of work on farms using SFSCs.

Furthermore, the results obtained are highly heterogeneous, depending among other factors on labor categories, sales channels, production practices and territories. More case studies are therefore desirable in order to better inform and explain this diversity, both within a territory and between different ones.

Since farms in SFSCs have a work organization as well as production and sales practices that vary over time, diachronic approaches should be used to account for changes in economic performance indicators over time, in order to better understand the effectiveness of organizational reconfigurations on income or labor productivity, for example.

### 9.4. Work organization

While the organization of work at the farm level has been the subject of numerous publications, none of them provides concrete information on the spatio-temporal linkages of production, processing and marketing tasks. Such approaches will be useful in better identifying the organizations adopted by farmers to cope with weekly work peaks (sales days) or seasonal ones (production peaks, harvests, etc.), and the spatial dynamics of SFSC-related work. The technical links between the choice of marketing channels and production practices also merit further exploration.

Since work is subject to frequent reorganization, diachronic methods will enable us to gain a better understanding of the causes and long-term consequences of these reorganizations. They will also provide us with more detailed information on the development of marketing channels and the farmers' successive learning processes.

Although fairly well covered in our corpus, the social relationships of production between different categories of workers deserve to be explored in greater detail, especially in the case of exclusively family collectives and those relying on salaried workers.

The pooling of tasks among producers and the delegation of certain tasks to territorial intermediaries have been identified as strategies to reduce workloads. As a result, these two processes too would benefit from further study from the perspective of territorial organization of work and would lead to a better understanding of the working relationships between various actors. Finally, the impact of territorial development actions and policies on work deserves to be studied in greater detail.

### 9.5. Working conditions

The theme of job satisfaction has been widely studied in various configurations. While these studies tend to focus on the farmer, other categories of workers – especially salaried ones – could be the subject of more in-depth studies. Even though the increased number of links

between SFSC actors is a source of job satisfaction, the relationships established with SFSC intermediaries deserve more exploration. Finally, diachronic approaches could be considered to better understand the links between organization, motivation and job satisfaction in an evolutionary perspective.

In contrast, the issue of physical and mental hardship associated with SFSC-related work has received very little attention. Given that there is a high risk in the agricultural sector of work-related accidents, overwork and even suicide, the impact of the increased complexity of work arising from SFSC participation on physical and mental stress deserves study. Such research would also make it possible to identify the categories of workers in SFSCs potentially most exposed to such risks, and to identify the factors contributing to this stress.

### 9.6. Occupation

Compared with other dimensions of work, the occupational dimension is largely understudied, except in terms of knowledge acquired and skills learned. Since marketing and production practices may differ from those of farmers who are not part of SFSCs, conceptions of the farming profession, and occupational norms and identities are likely to evolve, and should be documented. As the forms of farming that use SFSCs are very diverse – as are the profiles of the people involved in SFSCs – conceptions of the farming profession are likely to vary. Such studies are already available in the non-English literature [123].

## 10.  Conclusion

The issue of farmers' work in SFSCs is the subject of limited interest in the English-language scientific literature. Even though the results are based on highly contrasting case studies, they are often convergent.

Due to their low numbers of intermediaries, SFSCs are often praised for allowing farmers to capture a greater added value. However, this added value's impact on farmer incomes is highly uneven. Furthermore, it is obtained at the cost of a large amount of often poorly remunerated work. This additional work, in processing and marketing tasks as well as in production, requires an organization of tasks that is made more complex by their diversity and the range of skills associated with them. To save time, this work is frequently reorganized, and the additional workload is handled primarily by family workers, as also by salaried employees, precarious workers (apprentices, trainees, seasonal migrants), consumers and other volunteers. Territorial and State actors sometimes provide support in the form of investment subsidies or technical assistance. While managing this workload can be stressful, the complexity of farming in SFSCs and the diversity of the tasks involved nevertheless contribute to the farmer's job satisfaction.

Thanks to various registers of proximity (spatial, social, organizational), SFSCs also bring consumers and producers closer together. This proximity to consumers is a major factor in work satisfaction for farmers, and a source of motivation for those considering marketing via SFSCs. Counterintuitively, proximity to consumers does not ensure that farmers have increased decision-making and organizational autonomy [126]. While they are partly freed from the constraints imposed by actors of long supply chains, farmers must nevertheless adapt their practices, reorganize their work and adjust selling prices to meet consumer expectations.

From a methodological point of view, the choice of a systematic literature review presents both advantages and disadvantages. Its main limitation lies in the potential exclusion of articles that explore the theme of work without using the terms of our thesaurus. Additional results might also have emerged if we had extended our query to other languages or other databases. That said, this method enabled us to summarize results from a wide range of case

studies. It provides a good overview of the diversity of fields of study that address the work of farmers using SFSCs, and this in a way that is both transparent and reproducible. This overview could be supplemented by a narrative review, whose complementary approach would no doubt provide further elements of understanding [33]. Given the rise of the topic of agricultural work over the last decade [119] and with marketing practices that use SFSCs becoming more widespread, a regular update of this systematic review will be necessary in order to keep the results up to date. Finally, this literature review proposes a research agenda around SFSC-related work of farmers.

SFSCs offer an alternative to long supply chains and embody the hope of a renewal not only of food systems, but also of agricultural working practices. Being part of this change is a significant source of motivation, both for farmers and for other categories of workers. Many have a vision of work that goes beyond remuneration and social security to include expectations of a relationship with oneself, with other individuals and with the living world. While this hope for a transition in food systems is promoted in the discourse of the State and territorial actors, it remains insufficiently supported in practice by them, despite the public policies adopted and actions undertaken at the territorial and national levels for the reterritorialization of agriculture and food systems. The desired transition of our food systems therefore rests heavily on the shoulders of farmers, farmer groups, and other voluntary and salaried workers who accept to take on the extra work involved in this transition at their own cost in pursuit of an ideal lifestyle and food system. This observation calls into question this model's capacity to grow and establish itself as a sustainable alternative to conventional marketing models, in an economic framework that has been identified as unfavorable to the development of SFSCs [127].

## Supporting information

**S1 Document. Dupé P, Dedieu B, Gasselin P, Ollivier G.** The work of farmers marketing via Short Food Supply Chains: Protocol for a Systematic Literature Review. 2023. doi. org/10.17180/NWMX-3X92.
(DOCX)

**S2 Table. Full-text records included and records excluded, with reasons for exclusion.**
(DOCX)

**S3 Table. Main references cited by the study corpus (n = 28, times cited > 4).**
(DOCX)

**S4 Table. Methods and assessment of risk of bias for each included study.**
(XLSX)

**S5 Table. PRISMA checklist.**
(DOCX)

**S6 Table. Data extracted from the included studies.**
(XLSX)

**S7 Table. List of articles from query output, after duplicate removal (n = 494).**
(DOCX)

## Acknowledgments

The authors are grateful to Kim Agrawal for the quality of his English translation of this text from the original French.

## Author contributions

**Conceptualization:** Philippine Dupé, Benoît Dedieu, Pierre Gasselin.

**Data curation:** Philippine Dupé, Benoît Dedieu, Pierre Gasselin, Guillaume Ollivier.

**Formal analysis:** Philippine Dupé.

**Funding acquisition:** Benoît Dedieu, Pierre Gasselin.

**Investigation:** Philippine Dupé.

**Methodology:** Philippine Dupé, Benoît Dedieu, Pierre Gasselin, Guillaume Ollivier.

**Project administration:** Philippine Dupé.

**Supervision:** Benoît Dedieu, Pierre Gasselin.

**Validation:** Philippine Dupé, Benoît Dedieu.

**Visualization:** Philippine Dupé.

**Writing – original draft:** Philippine Dupé.

**Writing – review & editing:** Philippine Dupé, Benoît Dedieu, Pierre Gasselin, Guillaume Ollivier.

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
