## [Decision Letter · Decision Letter 0]

21 Aug 2024

PONE-D-24-29719The work of farmers in Short Food Supply Chains: Systematic Literature Review and research agendaPLOS ONE

Dear Dr. Dupé,

Thank you for submitting your manuscript to PLOS ONE. After careful consideration, we feel that it has merit but does not fully meet PLOS ONE’s publication criteria as it currently stands. Therefore, we invite you to submit a revised version of the manuscript that addresses the points raised during the review process.

We look forward to receiving your revised manuscript.

Kind regards,

Niranjan Devkota, PhD

Academic Editor

PLOS ONE

Journal Requirements:

This work was financially supported by the Fondation de France and by the metaprogram "Métabio" from the National Research Institute for Agriculture, Food and the Environment (INRAE, France), which financed a PhD grant for Philippine Dupé. The funders had no role in study design, data collection and analysis, decision to publish, or preparation of the manuscript.

3. Please upload a new copy of Figures 2, 5, 6 and 7, as the detail is not clear. Please follow the link for more information: https://blogs.plos.org/plos/2019/06/looking-good-tips-for-creating-your-plos-figures-graphics/" https://blogs.plos.org/plos/2019/06/looking-good-tips-for-creating-your-plos-figures-graphics/"

**Additional Editor Comments:**

Dear Authors

The paper is well-written and provides sufficient details; however, there are still some areas that need to be addressed.

Both reviewers have suggested, and I agree, that the headings and sub-headings should be streamlined to better align with the context and adhere to standard conventions.

Additionally, the manuscript in its current form is lengthy, with too many subheadings. Condensing the content and reducing the number of subheadings would improve readability and focus. Hence with the help of provided sample copy, please reduce it.

I also noticed that the authors have cited only one paper published in 2024. Given the availability of numerous relevant papers on this topic, I recommend including more papers available for 2024 and beyond.

I would like to suggest following the guidelines provided by the journal.

To assist with the revisions, I have provided examples of Systematic Literature Reviews (with its link) previously published in PLOS ONE. Please check and follow any of the styles.

• Brownfield land and health: A systematic review of the literature https://journals.plos.org/plosone/article?id=10.1371/journal.pone.0289470

• Moral judgment and hormones: A systematic literature review https://journals.plos.org/plosone/article?id=10.1371/journal.pone.0265693

• Zoonoses in Veterinary Students: A Systematic Review of the Literature https://journals.plos.org/plosone/article?id=10.1371/journal.pone.0169534

• Children Reading to Dogs: A Systematic Review of the Literature https://journals.plos.org/plosone/article?id=10.1371/journal.pone.0149759

• Early warning systems in obstetrics: A systematic literature review https://journals.plos.org/plosone/article?id=10.1371/journal.pone.0217864

Reviewers' comments:

Reviewer's Responses to Questions

**Comments to the Author**

1. Is the manuscript technically sound, and do the data support the conclusions?

Reviewer #1: Yes

Reviewer #2: Yes

2. Has the statistical analysis been performed appropriately and rigorously? 

Reviewer #1: Yes

Reviewer #2: N/A

3. Have the authors made all data underlying the findings in their manuscript fully available?

Reviewer #1: Yes

Reviewer #2: Yes

4. Is the manuscript presented in an intelligible fashion and written in standard English?

Reviewer #1: Yes

Reviewer #2: Yes

5. Review Comments to the Author

**Reviewer #1: ** Reviewer comments 16th August 2024 same for editor and author is attached.

Overall, the manuscript is not publishable in its current form. I recommend MINOR REVISIONS for publication in the Journal after the author/s addresses the above queries and suggestions given.

**Reviewer #2:**  "The Work of Farmers in Short Food Supply Chains: Systematic Literature Review and Research Agenda" is well written. The strength of the manuscript I observed is its methodology, which covers the papers from the two largest databases with significant consideration taken while selecting the papers. The papers presents the results in five themes: 1) structural 27 elements of work; 2) economic performance of work; 3) organization of work; 4) working conditions; 28 5) occupation.

The paper would be more attractive if it developed a theoretical model to make the Short food supply chain more effective. Besides, it will contribute significantly if some policy review in the global south and north context is incorporated. Besides, the research implications are another important part that needs to be addressed. Moreover, the papers seem very lengthy; if possible, it is better to reduce the size.

6. PLOS authors have the option to publish the peer review history of their article (what does this mean? ). If published, this will include your full peer review and any attached files.

**Do you want your identity to be public for this peer review?** For information about this choice, including consent withdrawal, please see our Privacy Policy .

Reviewer #1: No

Reviewer #2: No

---

## [Author Response · Author response to Decision Letter 1]

5 Oct 2024

Dear Editor and Reviewers,

We sincerely appreciate your insightful comments and valuable suggestions regarding our manuscript. In response to your recommendations, we have undertaken a substantial update of our Systematic Literature Review. This process has resulted in the incorporation of 23 additional documents into our corpus, expanding the total number of analyzed articles from 56 to 79. These new references are cataloged in the table at the end of this document. This expansion has significantly enriched our review's findings while maintaining the coherence of our primary conclusions. Consequently, the results sections have been substantially enhanced. Furthermore, we have bolstered the introduction with key references from our field of research.

In an effort to improve readability, we have made considerable strides in increasing conciseness, including reducing the overall manuscript length and shortening section titles. We have also ensured strict adherence to the methodological requirements specific to Systematic Literature Reviews. We trust that these modifications meet your expectations and elevate the overall quality of our manuscript.

Below, we provide point-by-point responses to your specific remarks. We would like to point out that, unlike our article, this document has not been reviewed by an English translator.

We look forward to your feedback on these revisions,

Philippine Dupé, Pierre Gasselin, Benoît Dedieu and Guillaume Ollivier.

Editor comments

1. I also noticed that the authors have cited only one paper published in 2024. Given the availability of numerous relevant papers on this topic, I recommend including more papers available for 2024 and beyond.

In accordance with your recommendations, we have substantially updated our Systematic Literature Review. 23 additional documents have been incorporated, increasing the total number of analyzed articles from 56 to 79. 9 of these new references were obtained through an updated query, including articles published between 01/01/2024 and 31/08/2024. 14 articles were subsequently selected via our "citation chasing" stage, utilizing a corpus of cited references and those citing our corpus. Indeed, this corpus was itself updated through our revised query. All new references are listed in the table at the end of this document. This addition has considerably enriched the results of our review while maintaining the consistency of our main conclusions. The various results sections have been extensively reworked to incorporate these new findings.

2. Both reviewers have suggested, and I agree, that the headings and sub-headings should be streamlined to better align with the context and adhere to standard conventions. Additionally, the manuscript in its current form is lengthy, with too many subheadings. Condensing the content and reducing the number of subheadings would improve readability and focus. Hence with the help of provided sample copy, please reduce it.

We have undertaken a significant effort to improve concision while maintaining the integrity and richness of our analysis. The manuscript has been reduced by approximately 6,000 characters (including spaces), equivalent to nearly 1,000 words. The final text, from abstract to conclusion, now comprises approximately 12,000 words. This revised length is in line with the standards for literature reviews in our research field. Three examples of comparable reviews are provided below for reference. We believe this revision strikes a balance between comprehensive analysis and concise presentation, adhering to the expectations of our field while addressing the need for brevity.

1. Martin T, Gasselin P, Hostiou N, Feron G, Laurens L, Purseigle F, et al. Robots and transformations of work in farm: a systematic review of the literature and a research agenda. Agron Sustain Dev. 2022;42. doi:10.1007/s13593-022-00796-2

2. O’Connor G, Reis K, Desha C, Burkett I. Valuing farmers in transitions to more sustainable food systems: A systematic literature review of local food producers’ experiences and contributions in short food supply chains. Agric Hum Values. 2024 [cited 30 Aug 2024]. doi:10.1007/s10460-024-10601-3

3. Csordas A, Lengyel P, Fuzesi I. Who Prefers Regional Products? A Systematic Literature Review of Consumer Characteristics and Attitudes in Short Food Supply Chains. Sustainability. 2022;14: 8990. doi:10.3390/su14158990

We faced two significant imperatives that limited our ability to reduce the manuscript’s length more than we did. Indeed, the incorporation of 23 new articles into our results has required additional content and analysis. In addition, to ensure our work qualifies as a Systematic Literature Review, we needed to include several methodological elements as required by the PRISMA method (S5 Table). However, we have removed two non-essential figures (relocated to S4 Table and Figs) for greater conciseness.

In response to your recommendations, we have shortened our headings, now focusing on key words to enhance readability. This revision was inspired by the exemplar articles you suggested. Finally, we propose to maintain a three-levels heading structure, which appears to be permitted by the journal. This structure enhances reading fluidity, by allowing for shorter paragraphs and more clearly signaling the nature of information addressed in each section.

Comments from Reviewer #2

1. The paper would be more attractive if it developed a theoretical model to make the Short food supply chain more effective.

The primary objective of our Systematic Literature review is to provide a rigorous and comprehensive assessment of the available scientific knowledge on the topic of farmers' work in SFSCs. To achieve this goal, we have adopted a transparent and reproducible methodology, detailed in a research protocol (S1 Document). Based on our findings, we also propose a research agenda in our discussion section. Our choice to conduct a Systematic Literature Review does not appear to be compatible with developing a theoretical model to improve SFSCs efficiency. However, our results could serve as a foundation for future work on such models. In our conclusion, we emphasize the need for increased research on farmers' work in SFSCs.

2. Besides, it will contribute significantly if some policy review in the global south and north context is incorporated.

Conducting such a review appears crucial to enhance the effectiveness of SFSCs. However, in this article, we chose to focus on scientific knowledge. Nevertheless, we call for our findings to be taken into account when designing public policies around SFSCs in our conclusion, and emphasize the importance of strengthening our understanding of farmers' work in SFSCs with various methodologies.

3. Besides, the research implications are another important part that needs to be addressed.

Our discussion section (§9) proposes a research agenda that details all the "knowledge gaps" identified across the various dimensions of work in our analytical grid. We specifically highlight the less explored sub-themes and the less frequently used methodologies.

4. Moreover, the papers seem very lengthy; if possible, it is better to reduce the size.

Cf Response to editor’s comments 2

Comments from Reviewer #1

1. The language and grammar are NOT of an acceptable standard. The structure and layout of the paper is not logical. Acronyms & Abbreviations sections given in the paper need to be deleted and each of these acronyms need to be used inside the paper. The tables need to be short wherever possible only.

Our article has been translated by an experienced English-speaking translator who is accustomed to collaborate with researchers in our field. The currently revised version has been proofread by the same translator.

Our paper is composed of:

- a contextual introduction (§1);

- a methods section (§2), detailing the various steps of the Systematic Literature Review methodology, on the basis of the PRISMA checklist (S5 Table);

- six results sections (§3-8), corresponding to the five analytical prisms of our research question and to a bibliometric analysis of the papers included in our review;

- a discussion section (§9), proposing a detailed research agenda;

- a conclusion.

To enhance readability, we have significantly reduced the length of our titles, now focusing on key words to enhance readability.

Acronyms and abbreviations are explained within the paper.

We have included only the essential tables for our article to be recognized as a Systematic Literature Review and have moved two figures to the supporting information (S4 Document).

2. Introduction section should be supported by important pieces of literature on the issues discussed. All of the challenges discussed are fine but need to be summarized.

We have added five references on our introduction to strengthen our presentation of the key issues, while also reducing the overall length of the introduction for greater conciseness. Four of these references are widely cited within our corpus (S3 Table).

• Renting H, Marsden TK, Banks J. Understanding Alternative Food Networks: Exploring the Role of Short Food Supply Chains in Rural Development. Environ Plan Econ Space. 2003;35: 393–411. doi:10.1068/a3510

• Forssell S, Lankoski L. The sustainability promise of alternative food networks: an examination through “alternative” characteristics. Agric Hum Values. 2015;32: 63–75. doi:10.1007/s10460-014-9516-4

• O’Connor G, Reis K, Desha C, Burkett I. Valuing farmers in transitions to more sustainable food systems: A systematic literature review of local food producers’ experiences and contributions in short food supply chains. Agric Hum Values. 2024 [cited 30 Aug 2024]. doi:10.1007/s10460-024-10601-3

• Allen P, FitzSimmons M, Goodman M, Warner K. Shifting plates in the agrifood landscape: the tectonics of alternative agrifood initiatives in California. J Rural Stud. 2003;19: 61–75. doi:10.1016/S0743-0167(02)00047-5

• Marsden T, Banks J, Bristow G. Food Supply Chain Approaches: Exploring their Role in Rural Development. Sociol Rural. 2000;40: 424–438. doi:10.1111/1467-9523.00158

3. The methodology adopted for analyzing all these is not clear and needs to be written in detail as a section on what methodology and approach followed in the whole paper, why, and how done

We have adopted a Systematic Literature Review methodology as outlined by Munn et al. (2018). In paragraphs §2.1, 2.2, 2.3, 2.4, and 2.5, we have detailed the various steps of this approach in accordance with the requirements of the PRISMA checklist (see Table S5). A detailed protocol, made public in October 2023, further specifies our choices and is included in the supporting information (S1 Document). It is referenced in the main text. To improve clarity, we have reduced the length of the methodology section and merged two paragraphs.

4. Review of literature section: The paper only reviewed major policy documents that were published on this topic until now. The coverage of the literature is limited need to be discussed the main findings of key papers. Therefore, from the perspective of a literature review, the paper has some obvious shortages too, which should be overcome. At the end of the review of the literature section, it would be nice to have a small summary paragraph or two-three sentences, that may have discussed the fundamental linkage of these papers. For example, what mechanisms are emphasized in each or overall paper? How are those older and newer papers linked to each other, summarized? After all, by reading the review someone would expect a clear “summary line" that helps to understand the connection and contribution of these papers. This is the first thing that should be improved in the paper, so that will be clear to the reader.

The primary objective of our systematic literature review is to provide a rigorous and comprehensive assessment of the available scientific knowledge on the topic of farmers' work in SFSCs. In our review, we have considered peer-reviewed articles, book chapters, or scientific reviews indexed in Web of Science and Scopus that meet our selection criteria, as detailed in our PICO table (Table 2). 79 articles were selected following careful examination of 920 articles (Fig 1) retrieved from a detailed search query (which is described in Table 1) and a citation chasing stage (§2.3). The selection process is detailed in section 2.3.

To complete our corpus, we have updated our review here, adding approximately 23 documents to our studied collection.

Each results section begins with an introduction presenting the nature of the various articles mobilized. This introduction serves to establish connections between these articles (methods, topics addressed).

The lack of information on certain topics is also a result of the review, which thus proposes a research agenda (§9). This agenda specifically highlights the less explored sub-themes and the less frequently used methodologies.

5. The article is a little deficient in providing insights on policy and practices in other countries as well, that have adopted different policy challenges.

Public policies are one element of results among others concerning farmers' work in SFSCs. Information on public policies impacting work of farmers marketing via SFSCs has been collected and summarized in the results sections §4.3 and §6.2.3. It is particularly noteworthy that very few papers dealing with work in SFSCs examine the dimensions of public policies related to this work. We have therefore identified this knowledge gap and call for strengthening research on this topic (§9.1, lines 779-780 and §9.2, lines 802-803).

6. What specific policies need to be taken to be integrated? There has been quite a large amount of work done in this area both theoretical and empirical from different countries.

In this review, we have chosen to focus on assessing the available scientific knowledge and proposing a research agenda. The collected articles present few results on public policies related to work in SFSCs. Therefore, we cannot propose a public policy agenda based on this corpus. However, we are convinced that our results could be used to design public policies around SFSCs. We believe that a more in-depth analysis of existing public policies could be the subject of another paper, employing different methodologies (databases, queries) than those used in this article.

7. The paper is though supported by theoretical backdrops. It would have been more useful had the author discussed other countries' experiences before bringing in other experiences. Again, a literature review of the policy and practice aspects of other countries of the world would provide a comparative perspective to the study and add value to the study.

In the introduction, we draw on references from various geographical contexts. In the results section §3.2, we emphasize that “The study of agricultural work in SFSCs, with SFSCs designated as such, is mostly carried out by Western researchers focusing on Western contexts.” Only 14 out of 79 publications come from South America, Africa, or Asia, even though marketing via SFSCs takes place in all countries of the world. This is therefore a significant finding. We thus call for strengthening case studies in other territories to better inform current practices in other locations.

8. The paper could have discussed themes using ideas from existing literature which is missing like, what is the role of Govt interventions in this case? Possibly, a thorough literature review on the theoretical aspects would be useful and could add to the scholastic value of the article.

Like other literature reviews, we have chosen to focus the discussion on developing a research agenda, rather than discussing new ideas by mobilizing other existing literature. We have been constrained by the already substantial size of the article and by the imperative to update the review. The selection of 23 new articles from 237 new references, as well as their analysis and integration into our results, indeed required considerable work, completed in just a month and a half.

The vast majority of articles in the corpus offer empirical results on our topic: our results therefore focus on these aspects rather than on theor

---

## [Decision Letter · Decision Letter 1]

6 Nov 2024

The work of farmers in Short Food Supply Chains: Systematic Literature Review and research agenda

PONE-D-24-29719R1

Dear Dr. Dupé,

We’re pleased to inform you that your manuscript has been judged scientifically suitable for publication and will be formally accepted for publication once it meets all outstanding technical requirements.

Kind regards,

Niranjan Devkota, PhD

Academic Editor

PLOS ONE

Additional Editor Comments (optional):

Dear authors,

It is my pleasure to inform you that both the reviewers suggest to accept your paper in this level. I am following the reviewers suggestion and would like to recommend for acceptance of the paper.

Wish you good luck.

Regards,

Reviewers' comments:

Reviewer's Responses to Questions

**Comments to the Author**

1. If the authors have adequately addressed your comments raised in a previous round of review and you feel that this manuscript is now acceptable for publication, you may indicate that here to bypass the “Comments to the Author” section, enter your conflict of interest statement in the “Confidential to Editor” section, and submit your "Accept" recommendation.

Reviewer #1: All comments have been addressed

Reviewer #2: All comments have been addressed

2. Is the manuscript technically sound, and do the data support the conclusions?

Reviewer #1: Yes

Reviewer #2: Yes

3. Has the statistical analysis been performed appropriately and rigorously? 

Reviewer #1: Yes

Reviewer #2: Yes

4. Have the authors made all data underlying the findings in their manuscript fully available?

Reviewer #1: Yes

Reviewer #2: Yes

5. Is the manuscript presented in an intelligible fashion and written in standard English?

Reviewer #1: Yes

Reviewer #2: Yes

6. Review Comments to the Author

Reviewer #1: Overall, the manuscript is publishable in its current form. I recommend for publication in the Journal as the author/s addresses the all queries and as per suggestions given.

Reviewer #2: (No Response)

7. PLOS authors have the option to publish the peer review history of their article (what does this mean? ). If published, this will include your full peer review and any attached files.

**Do you want your identity to be public for this peer review?** For information about this choice, including consent withdrawal, please see our Privacy Policy .

Reviewer #1: No

Reviewer #2: No

---

## [Editor Report · Acceptance letter]

PONE-D-24-29719R1

PLOS ONE

Dear Dr. Dupé,

I'm pleased to inform you that your manuscript has been deemed suitable for publication in PLOS ONE. Congratulations! Your manuscript is now being handed over to our production team.

Kind regards,

on behalf of

Dr. Niranjan Devkota

Academic Editor

PLOS ONE